# Comparison of Exosomes Derived from Non- and Gamma-Irradiated Melanoma Cancer Cells as a Potential Antigenic and Immunogenic Source for Dendritic Cell-Based Immunotherapeutic Vaccine

**DOI:** 10.3390/vaccines8040699

**Published:** 2020-11-19

**Authors:** Woo Sik Kim, DaeSeong Choi, Ji Min Park, Ha-Yeon Song, Ho Seong Seo, Dong-Eun Lee, Eui-Baek Byun

**Affiliations:** 1Research Division for Radiation Science, Korea Atomic Energy Research Institute, Jeongeup 56212, Korea; kws6144@kaeri.re.kr (W.S.K.); dschoi@kaeri.re.kr (D.C.); songhy@kaeri.re.kr (H.-Y.S.); hoseongseo@kaeri.re.kr (H.S.S.); 2General Toxicology Research Center, Korea Institute of Toxicology, Jeongup 56212, Korea; Jimin.park@kitox.re.kr

**Keywords:** melanoma cancer exosome, gamma irradiation, dendritic cells, tumor antigen-specific multifunctional T cells, vaccine

## Abstract

Cancer cells can secrete exosomes under various stressful conditions, whose functions are involved in the delivery of various biologically active materials into host cells and/or modulation of host immune responses. Therefore, an improved understanding of the immunological interventions that stress-induced tumor exosomes have may provide novel therapeutic approaches and more effective vaccine designs. Here, we confirmed the phenotypical and functional alterations of dendritic cells (DCs), which act as a bridge between the innate and adaptive arms of immunity, following non-irradiated (N-exo) and gamma-irradiated melanoma cancer cell-derived exosome (G-exo) stimulation, and evaluated the N-exo- and G-exo-stimulated DCs as therapeutic cancer vaccine candidates. We demonstrated that G-exo-stimulated DCs result in DC maturation by the upregulation of surface molecule expression, pro-inflammatory cytokine release, and antigen-presenting ability, and the downregulation of endocytic capacity. In addition, these cells promoted T cell proliferation and the generation of T helper type 1 (Th1) and interferon (IFN)-γ-producing CD8^+^ T cells. However, N-exo-stimulated DCs induced semi-mature phenotypes and functions, eventually inhibiting T cell proliferation, decreasing IFN-γ, and increasing IL-10-producing CD4^+^ T cells. In addition, although N-exo and G-exo stimulations showed similar levels of antigen-specific IFN-γ production, which served as tumor antigen sources in melanoma-specific T cells, G-exo-stimulated DC vaccination conferred a stronger tumor growth inhibition than N-exo-stimulated DC vaccination; further, this was accompanied by a high frequency of tumor-specific, multifunctional effector T cells. These results suggest that gamma irradiation could provide important clues for designing and developing effective exosome vaccines that can induce strong immunogenicity, especially tumor-specific multifunctional T cell responses.

## 1. Introduction

Cancer cells have been reported to secrete exosomes (also known as 30–150 nm extracellular nanovesicles), which have a lipid bilayer structure [1]. Importantly, these exosomes could influence all stages (tumor growth, angiogenesis, and immune surveillance) of cancer progression, indicating that they could be a potential therapeutic target for various cancers [2,3]. In addition, they are reported to contain various biological constituents of cancer cells, such as proteins and nucleic acids (DNA and RNA) [4]. This indicates that they may have the antigenicity of cancer cells. Therefore, they may also be used as potential diagnostic and vaccine candidates that are studied based on antigenicity [5,6]. Furthermore, many researchers have reported the function of exosomes as delivery systems for various therapeutic agents [7,8]. Despite these advantages, their applications as vaccines, diagnostics, and delivery systems may not be properly utilized without fully understanding the properties and functions of exosomes. Thus, a better understanding of the functional characteristics of exosomes may facilitate a rational design for developing vaccines that are more effective, diagnostics, and delivery systems.

In fact, the biogenesis, composition, and secretion of exosomes from cancer cells can be affected by various stress and disease conditions [9]. It is worth noting that tumor-derived exosomes produced by external stress, such as heat shock, chemotherapeutic drugs, and irradiation, in lung carcinoma, hepatocellular carcinoma, and breast cancer, can induce various anti-cancer immune systems (activation of Th1, CD8^+^ T cells, and/or NK cells), which, in turn, are useful as immunotherapeutic vaccine candidates [10,11,12]. Alternatively, exosomes released under hypoxic conditions promote the migration, invasion, and proliferation of tumor cells, as demonstrated in several studies; these were caused by the differentiation of M2 macrophages that contribute to the cancer cells’ ability to evade anti-tumor immunity [13,14,15]. Considering this information, the immunological characterization of tumor-derived exosomes produced under various stresses may help us to understand how tumors evade the immune system and how exosomes manipulate the host immune response toward anti-tumor immune responses.

Since several studies have shown that exosomes produced in response to stressful conditions can play various roles (immune evasion and protection) in anti-cancer immunity, our study aimed to evaluate and compare the immunological functions and antigenicity of exosomes produced from gamma-irradiated cancer cells and normal cancer cells. In addition, to further elucidate the functions of the exosomes, the immunogenicity and potential protective effect of each exosome-stimulated dendritic cell (DC) were compared in a head-to-head manner in melanoma tumor-bearing mice.

## 2. Materials and Methods

### 2.1. Animals

Seven- to eight-week-old C57BL/6 and BALB/c female mice were purchased from Orient Bio Inc. (Seoul, Korea). All animal studies were performed according to guidelines approved by the Institutional Animal Care and Use Committee (IACUC) of the Korea Atomic Energy Research Institute (permit number: KAERI-IACUC-2019-001).

### 2.2. Isolation and Characterization of Melanoma Cancer Cell-Derived Exosome

For exosome isolation, “B16/BL6” (C57BL/6 background) murine melanoma cell lines were generously provided by the Korean Cell Line Bank (KCLB, Seoul, Korea). The cells were maintained in Dulbecco’s modified Eagle’s medium (DMEM; GIBCO, Carlsbad, CA, USA) containing 10% fetal bovine serum (FBS, GIBCO) and 1% penicillin/streptomycin (GIBCO) in a humidified incubator containing 5% CO_2_ at 37 °C. The cells were harvested and resuspended in X-Vivo 15 chemical-defined serum-free culture medium (Lonza, Inc., Allendale, NJ, USA). Afterwards, the cells (1 × 10^6^ cells/mL) were irradiated at a dose of 100 Gy by a Gammacell 40 exactor of a ^137^Cs source (MDS Nordion, Ottawa, ON, Canada). Both non-irradiated and irradiated cells were cultured in serum-free media for 24 h. The supernatants were collected, and the pellets were discarded, following centrifugation at 2000× *g* for 20 min at 4 °C to remove residual cells and debris, and then at 10,000× *g* for 50 min at 4 °C to remove microparticles, before passing through a 0.22 μm syringe filter (Corning, NY, USA). Exosomes were then centrifuged at 100,000× *g* for 150 min, and washed in cold PBS followed by another centrifugation step (100,000× *g*, 150 min). Exosomes were resuspended in cold PBS and filtered through a 0.22 μm syringe filter. The protein concentrations of the exosomes were measured using a BCA protein assay kit (Thermo Scientific, Waltham, MA, USA). For exosome characterization, exosomes were loaded on formvar-carbon-coated electron microscopy grids (EMS, Hatfield, PA, USA) and were negatively stained with a 2% uranyl oxalate solution (Sigma, St. Louis, MO, USA). Photos taken were visualized using a JEOL 100CX II transmission electron microscope (TEM; JEOL Ltd., Akishima, Japan). In addition, the size distribution of exosomes was analyzed by nanoparticle tracking analysis (NTA; System Biosciences, Palo Alto, CA, USA). Exosomal protein expression was confirmed using antibodies against exosome markers, such as CD63 (Santa Cruz, CA, USA), CD81 (Santa Cruz), TSG101 (Abcam, Cambridge, MA, USA), and a negative marker, GM130 (BD Bioscience, San Diego, CA, USA). Briefly, cells and exosomes (10 μg of proteins) were lysed using RIPA buffer (Pierce, Rockford, IL, USA), and immunoblotting of each protein was performed as previously described [16].

### 2.3. Cytotoxicity of Exosomes in Bone Marrow-Derived Dendritic Cells (BMDCs)

BMDCs were generated from bone marrow cells isolated from C57BL/6 mice, as previously described [17]. Day 8 immature BMDCs were incubated with lipopolysaccharide (LPS, 100 ng/mL; Invitrogen, San Diego, CA, USA) or exosomes (5 and 25 μg/mL) isolated from non-irradiated (N-exo) and gamma-irradiated cancer cells (G-exo) for 18 h in a humidified incubator containing 5% CO_2_ at 37 °C. Cells were then harvested and stained with Annexin V and PI (BD Bioscience) according to the systematic instructions provided by the manufacturer. Samples (stained cells) were analyzed using a FACS instrument (MACSQuant VYB; Miltenyi Biotec, San Diego, CA, USA).

### 2.4. Analysis of Exosomes Incorporated by BMDC

The exosomes (1 mg/mL) were stained with 1 μM CellTrace^TM^ carboxyfluorescein succinimidyl ester (CFSE) staining solution (Invitrogen, Carlsbad, CA, USA) for 10 min at 37 °C, and then washed with cold PBS at 100,000× *g* for 120 min at 4 °C. CFSE-labeled exosomes were resuspended in cold PBS. Next, BMDCs were incubated with exosomes (5 and 25 μg/mL) for various time periods (30, 60, 12, and 180 min). In each treatment condition, cells were stained using anti-CD11c and analyzed using a fluorescence-activating cell sorting (FACS) instrument (CFSE^+^ CD11c^+^ cells).

### 2.5. Analysis of High Mobility Group Box Protein 1 (HMGB1) Level in Exosomes

HMGB1 concentration was measured in the isolated exosomes (N-exo and G-exo) using a commercial mouse HMGB1 ELISA kit (Arigo Biolaboratories, Taiwan, ROC) according to the manufacturer’s protocol.

### 2.6. Analysis of DC Maturation Induced by Exosomes

BMDCs were incubated with LPS (100 ng/mL) and exosomes (5 and/or 25 μg/mL) for 18 h at 37 °C in the presence or absence of the GolgiPlug protein transport inhibitor (BD Bioscience). For surface molecule analysis, in the absence of GolgiPlug, cells were stained with V450-conjugated anti-CD11c (PE-Cy7-CD11c), APC-CD80, PE-Cy7-CD86, PE-MHC-I, and FITC-MHC-II antibodies (Abs) for 30 min at 4 °C. All antibodies were obtained from eBioscience (San Diego, CA, USA). Samples were analyzed using a FACS instrument. For extracellular cytokine analysis, in the absence of GolgiPlug, TNF-α, IL-12p70, and IL-10 cytokine levels were measured in the culture supernatant using a commercial ELISA kit (eBioscience) according to the manufacturer’s protocol. For intracellular cytokine analysis, in the presence of GolgiPlug, cells were stained with V450- CD11c Ab for 30 min at 4°C and then fixed and permeabilized with a Cytofix/Cytoperm kit (BD Bioscience) according to the manufacturer’s instructions. Next, cells were stained with fluorescein-conjugated cytokine Abs (APC-TNF-α, PE-IL-12p70, and FITC-IL-10) for 30 min. These cytokine Abs were obtained from eBioscience. Samples were analyzed using a FACS instrument. For antigen uptake ability, in the absence of GolgiPlug, cells were incubated in the presence of 1 mg/mL FITC- dextran (40,000 Da, Sigma) at 37 °C or 4 °C. After 30 min of treatment, cells were stained using PE-Cy7-CD11c Ab (eBioscience) and analyzed using a FACS instrument. Unlike the above experiments, antigen presenting ability was performed as previously described [18]. Briefly, cells were incubated with LPS (100 ng/mL) and exosomes (5 and 25 μg/mL) in the presence of the Eα_44–76_ peptide (20 μg/mL; AbFrontier, Seoul, Korea) or OVA protein (250 μg/mL; Sigma) to analyze peptide-MHC-I-complex and peptide-MHC-II-complex formation. After 18 h of incubation, each cell was stained with PE-Cy7-CD11c, anti-25-D1.16 (eBioscience), or anti-Y-Ae Abs (eBioscience) for 30 min at 4 °C. As positive controls, cells were treated with 1 μg/mL OVA_257–264_ (AbFrontier; MHC-I) or 0.5 μg/mL Eα_52–68_ (AbFrontier; MHC-II) for 3 h. Samples were analyzed using a FACS instrument.

### 2.7. Mixed Lymphocyte Reaction (MLR)

The spleens of BALB/C mice were filtered through a cell strainer (40 µm, BD Biosciences) in DMEM complete medium (containing 2% FBS). Red blood cells (RBCs) present in the filtrate were lysed for 5 min with RBC lysis buffer (Sigma) and washed with DMEM complete medium. The CD4^+^ and CD8^+^ T cells from single-cell suspensions for spleen cells were separated by bead-conjugated anti-CD4 and -CD8 Abs (Miltenyi Biotec, San Diego, CA) according to the systematic instructions provided by the manufacturer. Isolated T cells were stained with 5 μM CFSE for 10 min at 37 °C, and then washed with PBS (supplemented with 2% FBS). Next, non-, LPS-, and exosome-treated DCs (treated for 18 h) were incubated with CFSE-labeled T cells at DC-to-T cell ratios (0.25:1 and 0.5:1). After 3 days of co-culture, cells were harvested and stained with V450-CD4 (eBioscience) and PE-Cy7-CD8 (eBioscience) Abs. Samples were analyzed using a FACS instrument. IFN-γ, IL-2, and IL-4 production was analyzed in culture supernatants using ELISA (eBioscience).

### 2.8. Antigenicity of Exosomes from Melanoma Tumor-Bearing Mice

B16/BL6 cells (5 × 10^5^ cells) were injected subcutaneously into the backs of C57BL/6 mice. Spleen cells collected from mice at 18 days post-injection were incubated with N-exo or G-exo at a concentration of 25 μg/mL. As a negative control, spleen cells collected from naive C57BL/6 mice were incubated with N-exo or G-exo. After 24 h of treatment, IFN-γ levels were evaluated in culture supernatants via ELISA.

### 2.9. Antitumor Effect and Tumor Antigen-Specific Multifunctional T Cell Induction by Exosomal Vaccination in Melanoma Tumor-Bearing Mice

B16/BL6 cells (5 × 10^5^ cells per site) were injected subcutaneously into the backs of C57BL/6 mice (7 mice per group). After three days of tumor cell injection, mice were vaccinated intravenously three times at three-day intervals with PBS only, immature DCs (iDCs), N-exo-loaded DCs, and G-exo-loaded DCs. Tumor size was measured every three days using the formula: area = (length × width^2^)/2. For analysis of the production of tumor antigen-specific multifunctional T cells, the spleens of each vaccination group (PBS only, iDCs, N-exo-loaded DCs, and G-exo-loaded DCs) 18 days post-tumor injection were filtered through a cell strainer (40 µm) in RPMI 1640 (GIBCO, Carlsbad, CA, USA) complete medium (containing 2% FBS), and then cells were lysed for 5 min with RBC lysis buffer and washed with RPMI 1640 complete medium. Next, spleen cells (2 × 10^6^ cells) were stimulated with tumor cell lysates (10 μg/mL) prepared from B16/BL6 cells for 12 h at 37 °C in the presence of GolgiPlug and GolgiStop protein transport inhibitors (eBioscience). B16/BL6 lysates used as a stimulant for spleen cells were generated by sonicating cancer cells, and protein concentration was quantified using a BCA protein assay kit. The stimulated cells were first blocked with Fc Block (anti-CD16/32; BD Bioscience) for 15 min at 4 °C, and then stained with Live/Dead cell staining kit (Invitrogen, Life Technologies, Naerum, Denmark), APC-Cy7-CD3 (eBioscience), FITC-CD4 (eBioscience), and Percp-Cy5.5-CD8 (eBioscience) Abs for 30 min at 4 °C. Next, cells were fixed and permeabilized with a Cytofix/Cytoperm kit. Finally, cells were stained with fluorescein-conjugated cytokine Abs (APC-TNF-α; BD Bioscience, PE-IFN-γ; BD Bioscience, and PE-Cy7-IL-2; eBioscience) for 30 min at 4 °C. Antigen-specific multifunctional T cells were analyzed using a FACS instrument (FACSverse, BD Bioscience).

### 2.10. Statistical Analysis

Analysis of variance between groups was performed using unpaired t-tests (between two groups) and one-way ANOVA (between three or more groups) using a statistical software (GraphPad Prism, version 7; San Diego, CA, USA). The results are expressed as the mean ± SD of three independent experiments. Values of * *p* < 0.05, ** *p* < 0.01, and *** *p* < 0.001 were considered statistically significant.

## 3. Results

### 3.1. Cellular Incorporation, Cell Viability, and Immunostimulant Potential of Tumor Cell-Derived Exosomes in Dendritic Cells

Tumor cells can induce exosomes to elicit immunogenicity changes under various stressful conditions, including irradiation [10,11,12]. Thus, analysis of the functional properties of these exosomes may help in understanding cancer pathogenesis and identifying novel exosomes, which can be potential vaccines. In order to study how exosomes derived from non-irradiated and gamma-irradiated melanoma cancer cells affect immune systems, particularly the innate and adaptive immune responses. We first purified melanoma cell-derived exosomes under non-irradiated (N-exo) and gamma-irradiated (G-exo) conditions. The sizes and concentrations of the purified exosomes were analyzed by the NTA system and their corresponding TEM images (Figure 1A,B). Results have shown that the size distribution (mean: 134.5 nm) and morphology of G-exo were similar to that of N-exo (mean: 129.1 nm), while the protein content was higher (*p* < 0.01) in G-exo than in N-exo. In addition, the levels of CD81, TSG101, and CD63 proteins used as classical exosome markers (However, a nonclassical exosomal marker, GM130, was detected in the cells.) were analyzed by western blotting. Cell lysates were used as negative markers. The results revealed that both G-exo and N-exo expressed CD81, TSG101, and CD63 proteins, confirming that these particles were exosomes (Figure 1C). These results suggest that gamma irradiation increases exosome production in melanoma cancer cells. Prior to confirming the immunological contributions of exosomes, the viability of BMDCs exposed to exosomes was analyzed to confirm whether G-exo and N-exo have cytotoxic effects on normal cells (Figure 1D). At concentrations (5 and 25 μg/mL) sufficient to induce various immune responses, G-exo and N-exo treatment showed no noticeable cytotoxic effect on BMDCs when measured with Annexin V/PI staining. We next confirmed the internalization of CFSE-labeled exosomes (G-exo and N-exo) by BMDCs. After incubation with BMDCs for various time periods (30 to 180 min), no significant difference was observed in the cellular uptake kinetics (CFSE-labeled exosomes^+^CD11c^+^) following the 5 μg/mL treatment concentration. Interestingly, at 25 μg/mL of exosomes, more G-exo cells were observed to have entered into the DCs comparing with N-exo at incubation periods of 60 and 120 min, which could be attributed to easier binding of G-exo to the DCs (Figure 1E).

Next, we confirmed whether exosomes derived from gamma-irradiated stress condition includes nuclear protein HMGB1, which is recognized as a danger associated pattern (DAMP) molecule, because tumor-derived HMGB1 released by heat shock stress condition has the potential to promote immune cell activation [19]. As shown in Figure 1F, the levels of HMGB1 were significantly higher (*p* < 0.05 in 5 μg/mL concentration, *p* < 0.01 in 25 μg/mL concentration) in exosomes (G-exo) derived from gamma-irradiated stress condition than in exosomes (N-exo) derived from non-irradiated condition. These results predicted that G-exo demonstrates an immunostimulatory effect and may induce the activation of innate and adaptive immune systems.

### 3.2. Comparison of Dendritic Cell Stimulatory Capacity of N-Exo and G-Exo

First, to examine the potential immunological roles of N-exo and G-exo, we investigated the differences in the immunocompetence between DCs treated with N-exo and G-exo, which is a critical phenotype in cross-presenting tumor antigens and generating effective anti-tumor T cell immunity [20]. In the present study, the analysis of immunological differences has focused on the phenotypic and functional changes induced by exosome stimulation on BMDCs, including: (1) the patterns of surface molecule expression and cytokine production based on the interactions of exosomes with the DCs, and (2) the DCs’ antigen uptake and presentation abilities. These analyses are essential for confirming DC maturation stages. We found that G-exo-stimulated DCs significantly augmented the expression of surface molecules (*p* < 0.001 for CD80, CD86, MHC-I and MHC-II with 5 and 25 μg/mL treatments) in a dose-dependent manner, while N-exo-stimulated DCs had similar or slightly higher expression levels (*p* < 0.01 for only MHC-I and MHC-II in treatment with 25 μg/mL) compared with non-treated DCs (Figure 2A). When extracellular (Figure 2B) and intracellular cytokine levels (Figure 2C) elicited by N-exo- and G-exo-stimulated DCs were analyzed, G-exo-stimulated DCs secreted significantly higher levels (*p* < 0.001 for 25 μg/mL) of the pro-inflammatory cytokines TNF-α and IL-12p70, which are essential cytokines for the differentiation of Th1 cells with anti-cancer T cell immunity [21] but not that of the anti-inflammatory cytokine IL-10. Importantly, N-exo-stimulated DCs showed a slight increase only in TNF-α production. We next confirmed the antigen uptake ability of exosome-stimulated DCs because matured DCs were previously shown to have decreased antigen uptake capacity [22]. As shown in Figure 1D, G-exo-stimulated DCs showed a greater decrease in dextran uptake ability (*p* < 0.001) than non-stimulated DCs (immature DCs), but N-exo-stimulated DCs showed only a slight decrease (*p* < 0.05). We further confirmed that exosomes can regulate the antigen-presenting ability of DCs. We found that G-exo-stimulated DCs exhibited significantly higher levels (*p* < 0.001) of antigen presenting ability for MHC-I and MHC-II compared with non-treated DCs; however, N-exo-stimulated DCs did not show any reaction (Figure 2E).

### 3.3. Comparative Analysis of T Cell Proliferation and Polarization Elicited by N-Exo- and G-Exo-Stimulated DCs upon Interacting with Naive T Cells

Next, to precisely characterize the role of exosomes in DC-T cell interactions, we investigated the immunocompetence differences (Figure 3A; T cell proliferation, Figure 3B,C; cytokine production) of N-exo- and G-exo-stimulated DCs upon interacting with naive CD4^+^ and CD8^+^ T cells. In allogenic MLR studies, G-exo-stimulated DCs, when interacting with T cells, promoted CD4^+^ and CD8^+^ T cell proliferation and augmented IFN-γ and IL-2 release compared with non-stimulated DCs; exposure to N-exo-stimulated DCs, on the other hand, showed decreased CD4^+^ and CD8^+^ T cell proliferation along with inhibited IFN-γ and increased IL-10 production (in CD4^+^ T cells only). However, there were no significant differences in the levels of IL-4 production between the groups. Here, IFN-γ and IL-2 are characteristic cytokines produced by Th1-type CD4^+^ T cells and activated CD8^+^ T cells, and IL-4 is a cytokine produced by Th2 cells [23]. IL-10 can directly inhibit the effector function of T cells. Interestingly, CD4 T cells that release this cytokine are known as IL-10-producing CD4 regulatory T (Tr1) cells, which have immunosuppressive functions [24,25]. Collectively, our results show that G-exo phenotypically and functionally activates DCs, and these cells can induce T cell proliferation, Th1 polarization, and activated CD8 T cell responses. On the other hand, N-exo-stimulated DCs have similar semi-mature phenotypes and functions (inhibited T cell proliferation and Tr1 cell induction), as seen in cancer environments [26,27,28].

### 3.4. Protective Effect and Multifunctional T Cell Responses Induced by N-Exo- and G-Exo-Stimulated DC-Based Vaccines in Tumor-Bearing Mice

The above results regarding DC maturation and T cell immunity induced by N-exo and G-exo motivated us to test the difference in immunogenicity and vaccine efficacy between N-exo- and G-exo-stimulated DCs in tumor-bearing models. Prior to evaluating the efficacy of exosome-based DC vaccines, we analyzed the antigen-specific IFN-γ response that was induced following treatment with N-exo- and G-exo in spleen cells isolated from non (normal)- and melanoma cell-injected mice. It is possible to predict whether exosomes have tumor antigenicity because these tumor antigens can generate memory T cell responses during cancer progression, and these memory T cells can promote strong antigen-specific IFN-γ responses [29,30]. As shown in Figure 4A, spleen cells isolated from melanoma cell-injected mice that were stimulated with N-exo- and G-exo showed a higher IFN-γ production compared to spleen cells of normal mice. Importantly, there were no significant differences between N-exo and G-exo. These results indicate that N-exo and G-exo have similar antigenicity against melanoma. We next investigated the protective efficacy and immunogenicity of N-exo- and G-exo-stimulated DC vaccination in tumor-bearing mice, as described in the Section 2. G-exo-stimulated DC-vaccinated mice showed reduced tumor growth compared with PBS- and non-stimulated DC-vaccinated groups (on the 12th day of tumor injection: *p* < 0.01, on the 15th and 18th day: *p* < 0.001). N-exo-stimulated DC-vaccinated mice also showed reduced tumor growth (on the 12th day: *p* < 0.05, on the 15th day: *p* < 0.01, on the 18th day: *p* < 0.001), but these protective effects were significantly higher (on the 15th and 18th day: *p* < 0.05) in G-exo-stimulated DC-vaccinated groups than in the N-exo-stimulated DC-vaccinated groups (Figure 4B). On the 18th day, when the difference in the protective effect between N-exo- and G-exo-stimulated DC vaccination was apparent, we evaluated the generation of tumor-reactive multifunctional T cells that can concurrently express multiple Th1 cytokines (including IFN-γ, TNF-α, and IL-2) using multicolor intracellular cytokine staining and flow cytometry (Figure 4C). These T cells play important roles in both initiating and maintaining effective anticancer immunity against various cancers [31,32]. As shown in Figure 4D,E, increased frequencies of activated CD44^+^CD4^+^ and CD44^+^CD8^+^ T cells (CD44 is an activation marker of T cells), which have multifunctional effector phenotypes, were observed in G-exo-stimulated DC-vaccinated groups, as compared to immature DC-vaccinated groups. In particular, triple cytokine-positive (IFN-γ^+^TNF-α^+^IL-2^+^ cells in CD4^+^CD44^+^ and CD8^+^CD44^+^ T cells) and double cytokine-positive (IFN-γ^+^TNF-α^+^, IFN-γ^+^IL-2^+^, TNF-α^+^IL-2^+^ cells in CD4^+^CD44^+^ T cells, and IFN-γ^+^TNF-α^+^, IFN-γ^+^IL-2^+^ cells in CD8^+^CD44^+^ T cells) antigen-specific T cells strongly increased. Importantly, these T cell responses were higher than those observed after N-exo-stimulated DC vaccination. These results show that G-exo-stimulated DC vaccination can induce more effective protection with a greater increase in the number of tumor-reactive multifunctional CD4^+^ and CD8^+^ T cells in tumor-bearing mice than in N-exo-stimulated DC vaccination. In addition, the higher vaccine efficacy of the G-exo-stimulated DC vaccination appears to be due to its more effective immunogenicity, because there is no difference in antigenicity between N-exo and G-exo.

## 4. Discussion

Understanding the immunological intervention of exosomes that contribute to the survival and development of tumors under various stressful and disease conditions is an important step for novel therapeutic approaches and effective vaccine design. Importantly, if enhanced activation of innate immune responses can help increase the adaptive immune response (especially T cell immunity) to a vaccine, weak innate immunity activation may not be as effective in protecting against cancers. Here, we investigated the different immunogenic properties of both exosomes (N-exo and G-exo) from non-irradiated and gamma-irradiated melanoma cancer cells upon interacting with DCs. Interestingly, we found that the G-exo moiety significantly enhanced DC maturation, along with the increased expression of surface molecules, pro-inflammatory cytokine TNF-α, and Th1-polarizing cytokine IL-12p70, and increased antigen-presenting ability, thereby promoting Th1 and activated CD8^+^ T cell responses. However, N-exo induced partial- or semi-mature DCs, and these N-exo-stimulated DCs promoted the inhibition of CD4^+^ and CD8^+^ T cell proliferation and IFN-γ expression produced by CD4^+^ T cells, and promoted the generation of IL-10-producing CD4^+^ T cells. These results indicate that N-exo may have less potential as a vaccine source compared to G-exo.

Recently, it has been reported that cancer cells exposed to gamma irradiation have immunostimulatory effects on the innate and adaptive immune responses [33,34,35]. Kim et al. showed that gamma (10 to 100 Gy)-irradiated colon cancer cells can enhance DC maturation when co-cultured with human dendritic cells, and these cells are apparently able to drive the differentiation of Th1 and cytotoxic CD8 T cells [33]. Strome et al. also reported that BMDCs loaded with irradiated E.G7 tumors (100 Gy), which express the OVA antigen, can promote the activation of the adaptive immunity by specifically generating OVA-specific IFN-γ-producing CD8^+^ T cells; treatment with these cells imparted significant protection in two analogous human melanoma models [34]. Similar to these experimental methods, the exosomes used in this study were isolated after 100 Gy irradiation. In fact, gamma-irradiated cancer cells induce the apoptotic cell death [36]. These cell death can enhance immune cell activation by increased expression of DAMP molecules, such as HMGB1 and heat shock proteins [19,37]. For example, Sanctis et al., showed that apoptotic tumor cell death induced by hyperthermia strongly improved DC maturation by secreting HMGB1 when co-cultured irradiated cancer cells and DCs [19]. This phenomenon (DC maturation via HMGB1 secretion) can also be induced by gamma irradiation [37]. Interestingly, our results showed that higher levels of HMGB1 were observed in exosomes (G-exo) derived from gamma-irradiated stress condition. In addition, gamma-irradiated melanoma cells released larger amounts of exosomes than non-irradiated cells, and these exosomes have immunogenic properties. Thus, these observations suggest that exosomes may be crucial components for promoting anti-cancer T cell immunity that is induced by DCs co-cultured with gamma-irradiated tumor cells.

Because exosomes, which are released from various cancer types, such as lung carcinoma and breast cancer, play a major role in suppressing anti-cancer T cell immunity by decreasing DC maturation, strategies to regulate their functions are important for cancer treatment [38,39]. However, when these exosomes are used as potential antigen sources in DC-based immunotherapeutic vaccines, they are known to induce strong anti-cancer immunity in the host [40,41]. Interestingly, Gu et al., reported that a DC vaccine loaded with tumor exosomes that induce an immunosuppressive effect in leukemia immunotherapy, led to prolonged survival and tumor growth inhibition, accompanied by an increase in tumor-specific CD4^+^ T cells in a mouse model with myeloid leukemia; however, these protective effects were not induced by tumor exosomes only [40]. Given this result, there is no difference in the antigenicity between exosomes of two types (e.g., exosomes derived under non-stress and stress conditions), but if there is a difference in immunogenicity, the vaccine efficacy of these exosome-loaded DCs could be different. Thus, the use of DCs in discovering new or improved exosomes and evaluating their vaccine efficacy can be a good strategy to assess both immunogenicity and antigenicity. In the present study, we found that in vitro G-exo stimulation produced a similar level of IFN-γ to that produced by N-exo stimulation in melanoma-specific T cells during the early phase of tumor growth, showing that both exosomes have similar antigenicity. Interestingly, the tumor growth in B16/BL6 melanoma-bearing mice was more efficiently delayed when mice were vaccinated with DC pulsed with G-exo than N-exo. Consequently, this vaccine effectiveness appears to be due to the difference in immunogenicity between N-exo and G-exo.

Among various lymphocyte subpopulations, CD4^+^ and CD8^+^ T cells with tumor-specific multifunctional effector phenotypes (co-expressing IFN-γ, TNF-α, and IL-2; these antigens are Th1-derived cytokines) are known to contribute to the activation of anti-cancer immunity, which leads to the expansion of memory and effector T cells [31,32,42]. In this study, an examination of DC-vaccinated mice showed that the G-exo-stimulated DC vaccine significantly induced tumor antigen-specific multifunctional CD4^+^ and CD8^+^ T cells in melanoma tumor-bearing mice. In particular, a higher frequency of these T cells was observed in the G-exo-stimulated DC vaccine than in the N-exo-stimulated DC vaccine. These data show that the protective efficacy in G-exo-stimulated DC vaccination correlated with the generation of tumor antigen-specific multifunctional CD4^+^ and CD8^+^ T cells and suggest that the use of the DC vaccine as an exosome delivery system is expected to be a potent strategy to supplement tumor exosome-mediated immunosuppression. Despite these results, future studies should further identify the immunogenic compounds, particularly other DAMP molecules, contained in G-exo and N-exo because this will help us understand how exosomes can promote immunogenic or non-immunogenic responses.

## 5. Conclusions

In conclusion, our study investigated the immunomodulatory properties and effects of exosomes derived from non-irradiated and gamma-irradiated melanoma cancers on innate and adaptive immunity, and the differences in DC vaccine efficacy for both exosomes in tumor-bearing mice. Although G-exo has the immunostimulatory capacity to induce T cell proliferation, Th1 polarization, and activated CD8^+^ T cell responses by stimulating the phenotypic and functional maturation of DCs, N-exo also has the capability to induce semi-maturation in DCs whose properties inhibit CD4^+^ and CD8^+^ T cell proliferation, and IFN-γ expression by CD4^+^ T cells; N-exo also promotes the generation of IL-10-producing CD4^+^ T cells. Interestingly, there was no difference in the antigenicity between these exosomes. In these situations, the G-exo-stimulated DC vaccine conferred protective multifunctional CD4^+^ and CD8^+^ T cell immunity and imparted significant protection to mice injected with B16/BL6 melanoma cancer cells. On the other hand, the N-exo-stimulated DC vaccine conferred a lower protective effect and multifunctional T cell immunity than the G-exo-stimulated DC vaccine. Taken together, immunogenic tumor exosomes induced by gamma irradiation are effective in forming a strong anti-cancer T cell immunity and inducing anti-cancer effects in subjects exposed to DC-based immunotherapeutic vaccines. 

## Figures and Tables

**Figure 1 vaccines-08-00699-f001:**
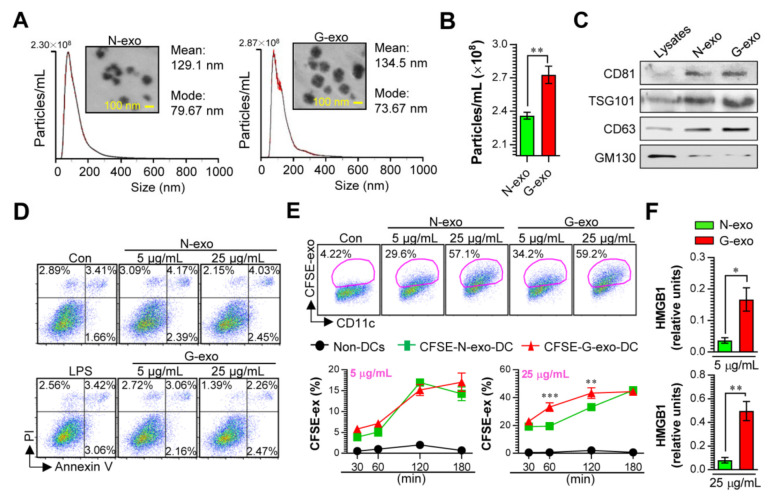
Analysis of the cytotoxicity, cellular uptake, and HMGB1 levels of N-exo and G-exo. Particle sizes (**A**; histograms) and numbers (**B**) of exosomes (N-exo and G-exo) purified from non- and gamma-irradiated B16/BL6 melanoma cancer cells were analyzed by the NTA system (*n* = 3 samples). (**A**; photo) TEM image of purified N-exo and G-exo. (**C**) The expression of exosomal markers such as CD81, TSG101, and CD63 were determined using western blot. GM130 is a negative marker of exosomes. (**D**) BMDCs were stimulated with LPS (100 ng/mL), N-exo (5 and 25 μg/mL) and G-exo (5 and 25 μg/mL) for 18 h, and respective cytotoxicities were measured using flow cytometry with Annexin V/PI staining (*n* = 3). (**E**) BMDCs were stimulated with CFSE-labeled N-exo and G-exo for indicated time periods and cellular uptake levels (CFSE^+^CD11c^+^ cells) of CFSE-labeled exosomes were analyzed by flow cytometry (*n* = 4 per time period). (**F**) HMGB1 levels were analyzed using ELISA in concentrations of 5 and 25 μg/mL of N-exo and G-exo (*n* = 3). * *p* < 0.05, ** *p* < 0.01, *** *p* < 0.001.

**Figure 2 vaccines-08-00699-f002:**
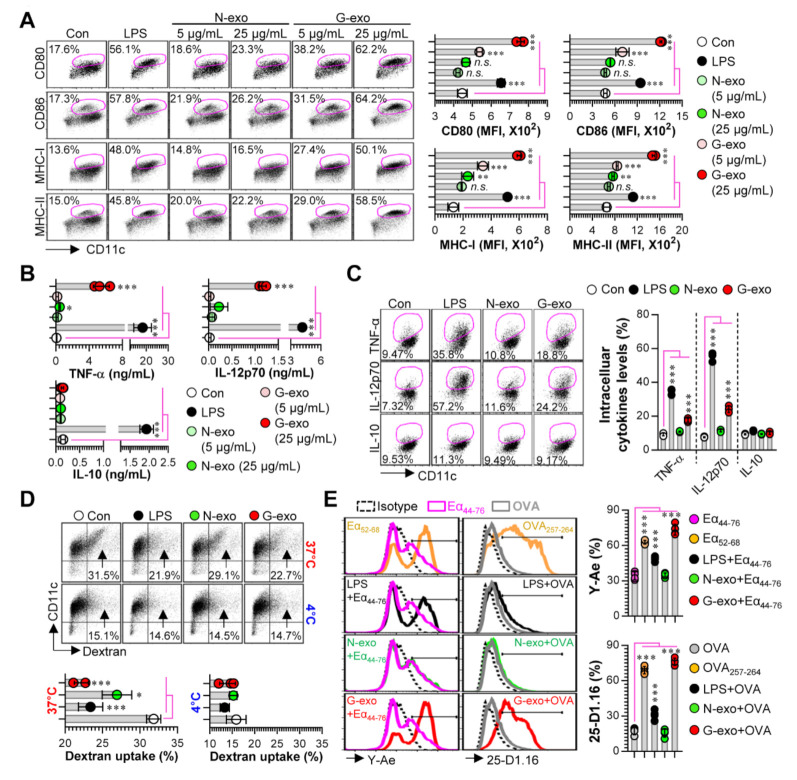
Analysis of surface molecules, cytokine production, antigen-uptake and presenting ability in N-exo- and G-exo-stimulated DCs. (**A**,**B**) BMDCs were stimulated with LPS (100 ng/mL), N-exo (5 and 25 μg/mL) and G-exo (5 and 25 μg/mL) for 18 h (*n* = 3). (**A**) The expression of surface molecules such as CD80, CD86, MHC-I, and MHC-II were analyzed using flow cytometry via their respective Ab staining. (**B**) TNF-α, IL-12p70, and IL-10 production were measured in culture supernatants via ELISA. (**C**) BMDCs were stimulated with N-exo (25 μg/mL) and G-exo (25 μg/mL) for 18 h in the presence of GolgiPlug, stained with anti-CD11c Ab, and were fixed/permeabilized. Cells were stained with anti-TNF-α, -IL-12p70, and -IL-10 Abs; the presence of the TNF-α^+^CD11c^+^, IL-12p70^+^CD11c^+^, and IL-10^+^CD11c^+^ cytokines were detected via flow cytometry (*n* = 3). (**D**) BMDCs were stimulated with N-exo (25 μg/mL) and G-exo (25 μg/mL) at 37 °C (positive condition) and 4 °C (negative condition). After 18 h, the cells were treated with Dextran-FITC for 30 min and analyzed for Dextran-FITC uptake levels (CD11c^+^Dextran^+^ cells indicate having an antigen-uptake ability) by flow cytometry (*n* = 3). (**E**) BMDCs were stimulated with LPS (100 ng/mL), N-exo (25 μg/mL), and G-exo (25 μg/mL) in the presence of OVA protein or Eα_44–75_ for 18 h and stained with anti-CD11c, -Y-Ae (in the Eα_44–75_ treatment condition), or -25-D1.16 (in the OVA treatment condition) Abs, and expression levels of Eα_44–75_/I-Ab and OVA_257–264_/H-2Kb complexes were analyzed in CD11c^+^ cells using flow cytometry (*n* = 4). Eα_44–75_- and OVA_257–264_-treated cells are positive control for MHC complexes. * *p* < 0.05, ** *p* < 0.01, *** *p* < 0.001. *n.s*.: not significant.

**Figure 3 vaccines-08-00699-f003:**
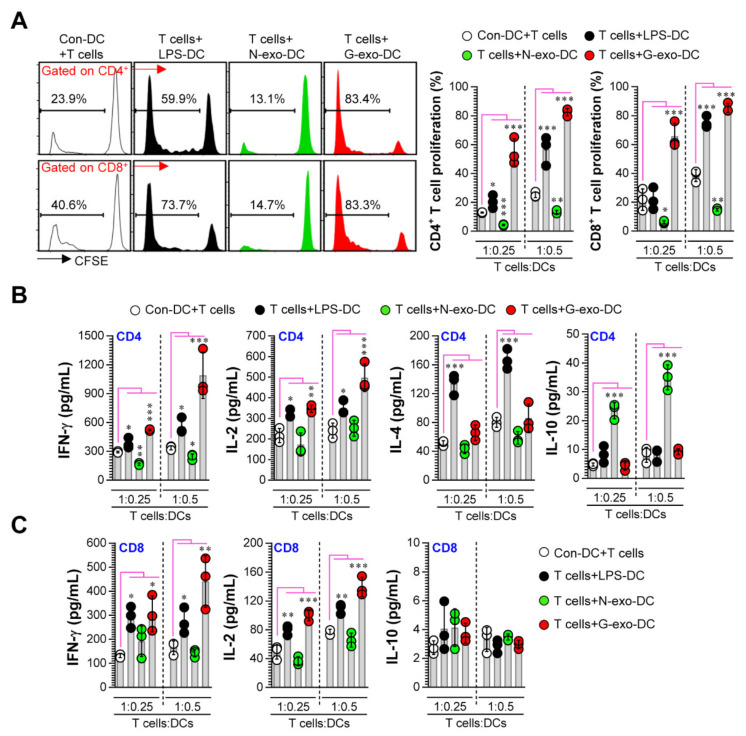
Analysis of proliferation and cytokine production of T cells induced by N-exo- and G-exo-stimulated DCs in the allogenic MLR assay. (**A**–**C**) Non-, LPS (100 ng/mL)-, N-exo (25 μg/mL)- and G-exo (25 μg/mL)-stimulated DCs incubated for 18 h were co-cultured with CFSE-labeled CD4^+^ and CD8^+^ T cells for 3 days (*n* = 3). (**A**) The proliferation of CD4^+^ and CD8^+^ T cells were analyzed using flow cytometry. (**B**) Production of IFN-γ, IL-2, IL-4, and IL-10 were analyzed in culture supernatants co-cultured with CD4^+^ T cells and DCs using ELISA. (**C**) Production of IFN-γ, IL-2, and IL-10 were analyzed in culture supernatants co-cultured with CD8^+^ T cells and DCs using ELISA. * *p* < 0.05, ** *p* < 0.01, *** *p* < 0.001. Con-DC: non-treated DC, LPS-DC: LPS-treated DC, N-exo-DC: N-exo-stimulated DC, G-exo-DC: G-exo-stimulated DC.

**Figure 4 vaccines-08-00699-f004:**
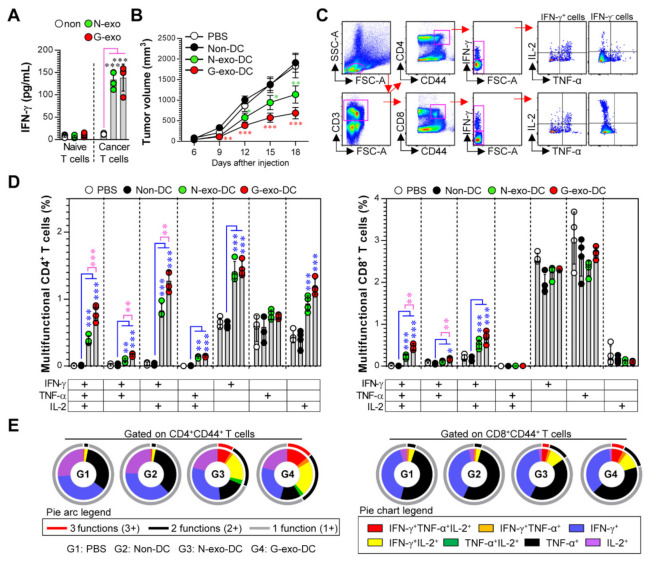
Analysis of antigenicity, immunogenicity, and vaccine efficacy of N-exo- and G-exo in tumor-bearing mice. (**A**) Spleen cells (*n* = 4) isolated from mice inoculated with melanoma cancer cells on the 18th day post-injection were treated with N-exo (25 μg/mL) and G-exo (25 μg/mL) for 24 h and IFN-γ secretion levels were analyzed using ELISA. (**B**–**E**) Mice inoculated with melanoma cancer cells were vaccinated three times at 3-day intervals with PBS only, immature DCs (Non-DCs), N-exo-loaded DCs (N-exo-DC), and G-exo-loaded DCs (G-exo-DC). (**B**) Tumor growth of vaccinated mice (*n* = 7 per group). One-way ANOVA test was used to evaluate significance. * *p* < 0.05, ** *p* < 0.01, *** *p* < 0.001, when compared to non-DC group. (**C**–**E**) On the 18th day after tumor inoculation, spleen cells from each group were isolated and stimulated with melanoma cell lysates (10 μg/mL) for 12 h in the presence of GolgiPlug and GolgiStop. Cells were then stained with T cell Abs and intracellular cytokine detection Abs as described in the Section 2 (*n* = 4 per group). (**C**) The gating strategy of tumor antigen-specific multifunctional CD4^+^ and CD8^+^ T cells. (**D**) The percentage of tumor antigen-specific multiple cytokine (IFN-γ, TNF-α, and/or IL-2)-expressing cells between splenic CD4^+^CD44^+^ and CD8^+^CD44^+^ T cells. (**E**) The pie charts present the mean frequencies of cells co-expressing IFN-γ, TNF-α, and/or IL-2. * *p* < 0.05, ** *p* < 0.01, *** *p* < 0.001. PBS: tumor only-injected mice, Non-DC: non-treated DC vaccination in tumor-bearing mice, N-exo-DC: N-exo-stimulated DC vaccination in tumor-bearing mice, G-exo-DC: G-exo-stimulated DC vaccination in tumor-bearing mice.

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
