# Peer review of "Comparison of Exosomes Derived from Non- and Gamma-Irradiated Melanoma Cancer Cells as a Potential Antigenic and Immunogenic Source for Dendritic Cell-Based Immunotherapeutic Vaccine"

_vaccines, 2020, doi:10.3390/vaccines8040699_

Round 1

Reviewer 1 Report

In the manuscript entitled “Comparison of exosomes derived from non- and gamma-irradiated melanoma cancer cells a potential antigenic and immunogenic source for dendritic cell-based immunotherapeutic vaccine” the authors demonstrated the immunostimulatory action of γ-irradiated tumor isolated exosomes on host immunity by inducing DCs maturation, antigen presentation, pro-inflammatory cytokines release. This effect resulted in higher T cell activation and proliferation. More importantly, a cancer vaccine based on DCs loaded with γ-irradiated tumor isolated exosomes was able to significantly reduce tumor growth in vivo. These results enlarge current understanding of anti-tumor vaccines and represent the ground for developing effective exosome-based therapeutics able to stimulate host immune system and to reactivate and potentiate tumor-specific multifunctional T cell responses. Well-designed experimental layout and proper manuscript writing support both authors’ conclusions and paper understanding. However, the authors need to improve the current version of the manuscript in order to achieve the publication by following the advices listed below.

MAJOR CONCERNS:

  • The authors elegantly prove the role of γ-irradiated melanoma-derived exosomes in activating DCs and supporting antigen presentation to T cells. However they nor demonstrate, either justify in discussion, possible mediators of this effect. Recent literature (DOI: 10.1016/j.vaccine.2018.05.010) demonstrates how tumor cells in stressed conditions can release DAMP (danger associated molecular patterns) molecules which in turn activate DCs in order to better perform antigen presentation and T cell activation. Could those molecules be stored also in the exosomes and carried to DCs? I would suggest the authors to evaluate whether HMGB1 or other DAMPs are involved in this process by performing an ELISA on γ-irradiated and normal tumor derived exosomes. Moreover, I would suggest adding in the discussion a small paragraph that puts into context possible mediators of DC activation, released by tumor cells in stressed conditions and that may be stored in the exosomes, mentioning what is already demonstrated with supernatant of treated tumor cells on DC activation as possible mechanism (DOI: 10.1016/j.vaccine.2018.05.010).

Minor issues:

  • Indicate in methods section how the FBS was exosome-depleted before use
  • Color codes in Fig 2E and Figure 4E are difficult to interpret. I warmly suggest revising them in order to support understandability to the reader. For the same reason I warmly suggest using consistent color code In figure 2 and 3 for Con, LPS, N-exo and G-Exo conditions.
  • In Figure 4B I suggest using ANOVA statistics instead of p value.

Reviewer 2 Report

To the author,

As a reviewer, I enjoyed reading this article. It is well-written logically, and the authors clarified the antigenicity of exosomes derived from gamma-irradiated melanoma cells, delineating another view of abscopal effect, which was supposed to occur through neoantigen released from the irradiated tumor. Research design is good, data are convincing, but I hope further experiments comparing the immunogenic compounds contained in G-exo and N-exo will be performed, hopefully in the next consecutive research at the authors’ lab.

I can suggest just a few minor points to improve this paper.

Minor concerns:

  1. The title of the paper is something wrong. Too long, but is it one-sentence?
  2. Please add some descriptions, regarding the isolation of spleen cells, in the Methods section.
